# Distinct Clades of Protein Phosphatase 2A Regulatory B’/B56 Subunits Engage in Different Physiological Processes

**DOI:** 10.3390/ijms241512255

**Published:** 2023-07-31

**Authors:** Behzad Heidari, Dugassa Nemie-Feyissa, Cathrine Lillo

**Affiliations:** IKBM, Department of Chemistry, Bioscience and Environmental Engineering, University of Stavanger, 4036 Stavanger, Norway; behzad.heidari60@yahoo.com (B.H.); dugassa.feyissa@uis.no (D.N.-F.)

**Keywords:** Arabidopsis, B56, brassinosteroid, cell cycle, flowering, PP1, PP2A, shugoshin

## Abstract

Protein phosphatase 2A (PP2A) is a strongly conserved and major protein phosphatase in all eukaryotes. The canonical PP2A complex consists of a catalytic (C), scaffolding (A), and regulatory (B) subunit. Plants have three groups of evolutionary distinct B subunits: B55, B’ (B56), and B’’. Here, the Arabidopsis B’ group is reviewed and compared with other eukaryotes. Members of the B’α/B’β clade are especially important for chromatid cohesion, and dephosphorylation of transcription factors that mediate brassinosteroid (BR) signaling in the nucleus. Other B’ subunits interact with proteins at the cell membrane to dampen BR signaling or harness immune responses. The transition from vegetative to reproductive phase is influenced differentially by distinct B’ subunits; B’α and B’β being of little importance, whereas others (B’γ, B’ζ, B’η, B’θ, B’κ) promote transition to flowering. Interestingly, the latter B’ subunits have three motifs in a conserved manner, i.e., two docking sites for protein phosphatase 1 (PP1), and a POLO consensus phosphorylation site between these motifs. This supports the view that a conserved PP1-PP2A dephosphorelay is important in a variety of signaling contexts throughout eukaryotes. A profound understanding of these regulators may help in designing future crops and understand environmental issues.

## 1. Introduction

Protein phosphorylation is an ancient regulatory mechanism involved in most cellular processes, and about half of the expressed proteins appear to become phosphorylated during their life time in Arabidopsis [1]. Phosphorylation status of a protein is the result of a balance between protein kinases and phosphatases. Generally, the number of genes encoding protein kinases is much higher than the number of genes encoding protein phosphatases. In Arabidopsis, there are about 1050 protein kinases balanced by only 150 catalytic protein phosphatases [2]. Due to the large number of regulatory subunits that are in a complex with catalytic subunits, the protein phosphatases become specific towards their substrates, and the number of protein phosphatases can match the number of protein kinases. Protein phosphatase 2A (PP2A) is a major serine/threonine protein phosphatase conserved in all eukaryotes. PP2A belongs to the PPP (phosphoprotein phosphatase) family of protein phosphatases, and PPPs are responsible for catalyzing about 90% of protein dephosphorylations in eukaryotes. PP2A is involved in a wide range of processes, including regulation of gene expression, hormone metabolism, cell cycle, tissue and organ development, metabolism, and responses to environmental signals including signals from microorganisms [3,4,5,6,7]. In mammals, PP2A is also well known as a tumor suppressor [8], and hence, it is of profound interest in medical research. PP2A enzymes have been much studied in yeast, animals, and higher plants. The catalytic PP2A subunits (C subunits) are found in the cell as a core heterodimeric complex with scaffolding subunits (A subunits), and as oligomeric complexes with various regulators, both activators and inhibitors [3,9,10]. The active PP2A complex generally consists of a catalytic C, a scaffolding A, and a regulatory B subunit. The B subunits belong to four different, evolutionary non-related families, B/B55, B’/B56, B’’/B72, and B’’’; the last group is not found in plants. Somewhat misleading, B is used as a collective term for all four groups, as well as for a specific group (B/B55). Here, we will use B as a collective term. Several investigations have shown that the B subunits confer substrate specificity and influence subcellular localization of the heterotrimeric PP2A complex. A basic survey of all the B subunits and their known physiological functions in Arabidopsis was recently published [11]. Important progress has been made concerning how members of the B’/B56 group can guide substrate specificity [12,13] and which processes are regulated by PP2A [7]. In this review, we look deeper into the characteristics and functions of the B’/B56 regulatory subunits, emphasize what has been found in the Arabidopsis model plant, and draw connections to yeast and mammalian PP2A-B’/B56.

## 2. B’ Structure and Recognition of Slim Motifs

As the name indicates, B’/B56 (from now on called B’) usually have a molecular weight of approximately 56 kDa; the Arabidopsis B’ subunits vary from 55 to 62 kDa. In humans, there are five B’ subunits (B’α/PP2AR5A, B’β/PP2AR5B, B’γ/PP2AR5C, B’δ/PP2AR5D, and B’ε/PP2AR5E). In *Saccharomyces cerevisiae*, there is only one B’, named RTS1, and *Saccharomyces pombe* has two B’ subunits, named Par1 and Par2. In Arabidopsis, there are as many as nine B’ subunits, and these can be divided into four clades which contain the members: (i) α, β, ε; (ii) γ, ζ; (iii) δ, η, θ; and (iv) κ (also called iota) (Table 1). A fifth clade, Φ, is also found in higher plants, but not in Arabidopsis, and appears to be involved in the interaction with mycorrhizal fungi. Arabidopsis and other Brassicaceae/Cleomaceae do not form mycorrhizal associations, and this could be the reason that B’ϕ has been lost from Arabidopsis [9,14,15]. The B’ clades evolved after the division of eukaryotes into animals and plants, and there is no connection between the use of Greek letters for animal and plant B’ subunits. Arabidopsis B’κ evolved 144 million years ago, and other B’ subunits evolved after emergence of flowering plants, 40–60 million years ago. Animal B’ isoforms diverged about 450 million years ago. The B’ proteins have a conserved core in all eukaryotes, but the carboxy and amino terminal ends vary in both length and sequence. Expansion of the B’ family was driven by functional diversification and positive selection [9].

The core region, the B56 domain, has eight HEAT (α-helical Huntington/Elongation factor 3/A subunit/TOR) repeats. Recently, HEAT repeat number four flanked by about seven amino acids from both neighboring HEAT repeats was shown to be important for recruiting specific substrates to mammalian PP2A [12]. These amino acids constitute a concave structure, a pocket, on the surface of the B’ protein with a hydrophobic patch and a basic patch conserved throughout eukaryotes, from yeast to mammals and plants. The pocket is important for binding SLiM motifs (short linear motifs), likely in all eukaryotes. SLiM motifs are short motifs, less than 10 amino acids, that are found in unstructured regions of proteins and usually bind with low affinity and transiently. A certain SLiM can often be found in many non-related proteins and bind to the same pocket on interacting proteins. This has led to the hypothesis that a SLiM formation is driven ex nihilo, i.e., “out of nothing”, and by acquiring a new SLiM, a protein could gain new functions [16]. SLiMs on several mammalian B’ substrates were investigated and found to be involved in binding to the B’ pocket [6,12]. Furthermore, proteins with SLiMs are not only recruited as a substrate to become dephosphorylated, but are also important for protein complex assembly, trafficking and anchoring to specific subcellular locations [12,16]. The SLiM interacting with the B’ pocket was identified as a LxxIxE motif. The leucine interacts with the hydrophobic patch in the pocket and the glutamate interacts with the basic patch [12]. Hertz and collaborators [12] undertook a bioinformatic analysis of different eukaryotes and listed (after filtering) 164 proteins containing the LxxIxE motif in *H. sapiens*, 41 in *S. cerevisiae* and 107 in Arabidopsis. These proteins are putative interacting proteins of B’, although other types of interactions between substrates and B’ also exist, as has for instance been found for shugoshin proteins in mammalian cells [17] or the PEST domain (proline-, glutamic acid-, serine-, and threonine-rich domain) in the transcription factor BZR1 in higher plants [18]. Analysis of the list of genes encoding LxxIxE motifs by the Panther classification tool [19] indicated a high number of proteins being involved in localization and vesicle processes. Genes annotated with the term “cellular localization” showed 40 genes and 1.85 times enrichment in *H. sapiens*. For Arabidopsis, this term showed 15 genes and 4.57 times enrichment. To extend our knowledge on B’ interacting proteins in Arabidopsis, proteins listed with the LxxIxE are potent candidates and should be further investigated to assess interactions. 

**Table 1 ijms-24-12255-t001:** Different clades of B’ regulatory subunits in Arabidopsis. Relative expression levels, number of predicted phosphorylation sites, and main subcellular localization for each B’ is presented. The nomenclature is according to locus identifier (AGI numbers) in the TAIR database and Booker and DeLong (2017) and Farkas et al. (2008) [9,20]. Expression data are RNAseq results for seedlings [21], and ATH1data for rosettes and pollen from the eFP Browser [22]. Number of predicted Ser/Thr phosphorylation hot spots are from the PhosPhAt Database (https://phosphat.uni-hohenheim.de/ accesed on 3 January 2023) [23]. Main subcellular localization is according to references given [24,25,26] and the SUBA database (https://suba.live/ accessed on 8 February 2023) [27].

Higher Plant B’ Clade	Each Clade Member Greek Letter/Number	AGI Locus Identifier and Protein UNIProtKB	Relative Expression in Seedlings%	Relative Expression in Rosette %	Relative Expression in Pollen %	Phosphorylation Predicted Number of Ser/Thr in Hot Spots	Matre 2009 Waadt 2015 Wang 2016 Main Localization (See Also Text)	SUBA Consensus Maximum Localization
α, β, ε/3,4,7	B’α/B3	At5g03470 O04375	21	52	20	2	nucleus	nucleus 1.0
	B’β/B4	At3g09880 O04376	42	45	15	7	nucleus	nucleus 1.0
	B’ε/B7	At3g54930 Q9SV41	9	18	73	4	cytosol nucleus	nucleus 0.76
γ, ζ/5,8	B’γ/B5	At4g15415 Q8RW96	39	37	16	0	cytosol nucleus	cytosol 0.958
	B’ζ/B8	At3g21650 Q9LVE2	21	65	6	11	cytosol mitochondrion nuclear	mitochondrion 0.728
δ, η, θ/6,9,10	B’δ/B6	At3G26030 Q9ZQY6	100	100	0.3	0	cytosol phragmoplast	golgi 1.000
	B’η/B9	At3g26020	36	32	43	15	cytosol nucleus	nucleus 1.000
	B’θ/B10	At1g13460	36	10	14	11	cytosol peroxisome nuclear	nucleus 0.496 golgi 0.406
κ/11	B’κ/B11	At5g25510 Q93YV6	70	76	100	11	cytosol nuclear	plastid 0.760 nucleus cytosol
ϕ	B’ϕ	not in Arabidopsis	-	-	-			

## 3. Expression Levels and Subcellular Localization Enable Diverse Functions of B’ Subunits

B’ transcripts are generally found throughout the plant, with some genes more highly expressed than others (Table 1). Strikingly, B’δ and B’ζ transcripts are present only at very low levels in pollen (Table 1), and this is also affirmed at the protein level according to the ATHENA database [1], though not as prominently as on the transcript level. Post-translational regulation of B’ subunits has not much been studied in plants but may also influence activities of the B’ proteins, as found in the cell cycle of fission yeast (*S. pombe*), where B’ was activated by dephosphorylation [28]. Phosphorylation of B’ in human can lead to both a decrease and increase in PP2A activity [29]. Conserved phosphorylation motifs in B’ were found in *S. pombe*, human (Grallert 2015) and the five Arabidopsis subunits, B’η, θ, ζ, γ, κ (details later, in Section 7.2). It was newly reported for Arabidopsis B55β that the phosphorylation state of B55β can be essential for forming an active PP2A-B55β complex. The B55β needed to be dephosphorylated to give an active complex [30]. According to the PhosPhat database, most Arabidopsis B’ subunits are predicted to have two or more serine and threonine residues, being hotspots for phosphorylation and making them interesting candidates for post-translational regulation to be further investigated (Table 1) [23,31]. 

Subcellular localization of PP2A regulatory subunits is important for the physiological functions of the different PP2A complexes in vivo, and the regulatory subunits are believed to direct PP2A complexes to their site of action (Figure 1). For examination of subcellular localization, recombinant genes encoding subunits with fluorescent protein-tags have been constructed and transformed into various types of cells for microscopic observations. However, various fusion combinations and tissues have been used, and obviously, this can give some contrasting results. For instance, a tag at the C-terminal end of the B’ subunit may hide a C-terminal signal peptide (for targeting to peroxisomes and secretion pathway), or likewise, an N-terminal peptide (for chloroplast and mitochondrion import) can be hidden by an N-terminal fluorescent protein. Reliable conclusions also depend on adequate expression levels, since cells with highly over-expressed proteins may give artefacts. An overview of the subcellular localization of Arabidopsis B’ subunits expressed in Arabidopsis or Nicotiana cells is shown in Table 1. B’a and B’b have mainly been found to localize to the nucleus, and in contrast to other B’ subunits, showed stable nuclear localization independent of N or C-terminal tag [25]. The nuclear localization is in agreement with their function in dephosphorylation of transcription factors [26] and involvement in chromatid binding [32]. Other subunits were found mainly to be cytosolic, such as B’γ [25,33] and B’δ [34]. Substrates for PP2A complexes containing B’γ and B’δ have not yet been identified, but their involvement in biological processes are suggested as a defense response and new cell wall formation in diving cells, respectively. Some B’ subunits were found in both nucleus and other compartments. The B’θ subunit was reported to localize with peroxisomes when the C-terminus was free, and it has a peroxisomal signal peptide 1 (PST1) at the C terminus. PP2A catalytic and scaffolding subunits could possibly enter peroxisomes as a complex together with B’θ [24,35], and/or B’θ can enable transfer of other proteins into peroxisomes as found for the SINA-like E3 ligase [36]. B’θ has also been reported to be nuclear [25]. 

Reversible protein phosphorylation is essential for regulation of various processes as well as inside chloroplasts and mitochondria. However, attempts to demonstrate PP2A (and PP1) activity in chloroplast preparations concluded that these activities were not found in chloroplasts [37,38]. Other types of protein phosphatases appear to be important in mitochondria and chloroplasts [39], including prokaryotic type Shewanella-like protein phosphatases [40,41]. According to the SUBA database, Arabidopsis PP2A catalytic subunits are generally localized to the plasma membrane, cytosol, and nucleus [27]; hence, these are also the main expected localizations for active B’ subunits. However, the B’ζ subunit colocalized with a mitochondrial marker when the N-terminal end was free [24]. B’ζ could possibly direct a complete PP2A complex to be targeted to the outer mitochondrial membrane and be engaged in dephosphorylation of proteins to be imported into the mitochondria. In human cells, both protein kinases and phosphatases, including PP2A, are known to be localized at the outer membrane of mitochondria, and these are regulating mitochondrial metabolism, fission, and apoptosis [42]. Several Arabidopsis proteins imported into mitochondria and possessing phosphorylation sites in the N-terminal sequence, become phosphorylated in the cytosol, but need dephosphorylation before import [43]. PP2A could be involved in dephosphorylation of such proteins to be imported, or alternatively, PP2A could be involved in dephosphorylating proteins in the outer mitochondrial membrane. The role of PP2A in preparation for import to various organelles is a subject deserving further studies in plants. 

## 4. PP2A Is Involved in Various Hormone Signaling Pathways

### 4.1. Auxin and Ethylene

One of the earliest findings on plant PP2A was its involvement in root growth. The *rcn1* (*roots curl in naphthylphthalamic acid 1*) mutant was found to have a mutation in one of the three PP2A scaffolding proteins (A1/RCN1) and revealed that PP2A was involved in regulating auxin transport [44]. The catalytic subunits C3 and C4 were found to dephosphorylate membrane-bound auxin efflux proteins (PINs). Recent work showed that root growth inhibition by ethylene implies PP2A activation. More specifically, the complex consisting of C4-A2-B55 was dephosphorylated on B55, thereby activated in the presence of ethylene, and this complex could dephosphorylate PIN2 (also called EIR1, ETHYLENE INSENSITIVE ROOT 1) [30]. The main focus has been on scaffolding and catalytic subunits, and little is known about B’ subunits in these processes [45,46,47], but as discussed in the review by Máthé et al. [11], PP2A-B’ζ binds to the essential ethylene signaling protein CTR1 (CONSTITUTIVE TRIPLE RESPONSE), stabilizes the protein, and inhibits ethylene signaling [48]. 

### 4.2. Brassinosteroid Signaling Is Regulated Positively by B’α and B’β and Negatively by Other B’ Subunits

The Arabidopsis B’α and B’β subunits are 83% identical on the protein level, and they function redundantly, because knocking out only one of the two corresponding genes still gives vigorous plants, whereas impairment of both genes results in abnormal plants [49]. The third member of the clade, B’ε, is 61% identical to B’α and its transcripts are present at a lower level than *B’α* and *B’β* transcripts in vegetative tissue, but higher in flowers. The B’ε subunit clearly does not replace the other two clade members to give healthy plants. The double mutant *b’αβ* (F1) still shows growth similar to WT, but self-fertilization of the homozygous double mutant resulted in variable progeny, many with severe dwarfism, and hardly any seeds. The phenotypes in the progeny occur because of defects in meiosis with resulting aneuploidy in following generations, and as a result of defects in BR signaling [18,32,49,50]. In Arabidopsis, the receptor kinase BRI1 (BRASSINOSTEROID-INSENSITIVE 1) at the plasma membrane is necessary for growth, and the receptor is phosphorylated and activated in response to binding of BR. PP2A complexes containing B’ subunits interact with BR signaling both by dephosphorylating the BRI1 receptor kinase (signal deactivation) and acting further downstream in the signal transduction chain by dephosphorylating the transcription factor BZR1 (BRASSINAZOLE-RESISTANT 1) (signal activation). Various in vitro experiments showed that several B’ subunits were candidates for participating in the dephosphorylation of both the BRI1 receptor and the BZR1 transcription factor. Immunoprecipitated PP2A complexes, including complexes with B’β and B’ε could dephosphorylate BRI1 in vitro [26]. However, in vitro, the substrate specificity is not necessarily high, and the catalytic subunits may be active towards many different substrates. BiFC (bimolecular fluorescence complementation) and blotting overlay experiments also pointed to protein–protein interactions between B’ from different clades with both the receptor kinase and the transcription factor [18,26]. In vivo experiments, on the other hand, with the dwarfed, weak *bri1-5* mutant transformed with various *B’* genes, were successfully used to sort out the different physiological roles of the B’ regulatory subunits. Overexpressing of B’α or B’β in the *bri1-5* mutant background had a positive effect on plant growth, and partially restored wild type phenotype [26]. This clearly underpins that these B’ subunits act positively and downstream of BR reception in the signaling pathway. Overexpression of B’ε had no effect [26]. Overexpressing of any of the other six B’ (γ, ζ, δ, η, θ, κ) had negative effects and resulted in even more severe dwarfism in the *bri1-5* mutant, Hence, B’α and B’β regulate BR signaling positively, while six other B’ subunits regulate BR signaling negatively. Further studies, revealing subcellular localization of the various B’ subunits, sorted out which B’ would likely interact with the receptor in the cytoplasm, excluding B’α and β from this function, but confirming B’α and B’β as regulators in the nucleus downstream in the BR signaling chain [26]. Although BRI1 and BZR1 are examples of substrates being dephosphorylated by PP2A-B’ complexes, they are not listed as containing the LxxIxE docking motif [12]. The reason for this can be other important binding motifs, such as the PEST motif in BZR1 [18], and the fact that active SLiMs can reside on other proteins that interact with the substrate, and hence, serve as scaffolding protein for PP2A-B’ and substrates, as found in mammalian cells [13].

### 4.3. Salicylic Acid Signaling—Involvement of B’γ

Salicylic acid has important functions in defense reactions towards pathogens and in plant growth and development. Salicylic acid influences auxin accumulation, distribution, and degradation. Simplified, low concentration of salicylic acid promotes growth, whereas high concentration slows growth and promotes defense reactions [51]. Salicylic acid was found to bind directly to PP2A scaffolding subunits A3 and A1/RCN1, leading to inhibition of PP2A and hyperphosphorylation of PIN2, which inhibited root growth [52]. Whether any specific B subunit is favored for these reactions was not investigated, but Durian and co-workers reported that PP2A-B’γ was involved in salicylic acid formation related to age-dependent leaf senescence as well as defense reactions [53]. 

## 5. Interactions with Microbes

PP2A plays a role as negative regulator of defense in plants [54]. This is likely part of a mechanism to antagonize protein kinases and thereby avoid unnecessary defense responses in plants when pathogens are not present. Silencing of PP2A catalytic subunits resulted in constitutive expression of defense genes and increased resistance to *Pseudomonas syringae* and *Rhizoctonia cerealis* in *Nicotiana benthamiana* and *Triticum aestivum*, respectively [54,55]. The first site to interact with microorganisms is on the cell surface of the host, and both plants and animals have PRRs (pattern recognition receptors) that recognize PAMPs/MAMPs (pathogen/microbe-associated molecular patterns). These receptors are usually part of larger complexes. For example, a receptor that recognizes flagellin in Arabidopsis, FLS2 (FLAGELLIN SENSING 2), forms a complex with the co-receptor BAK1 (BRASSINOSTEROID-ASSOCIATED KINASE 1) (Figure 1). When PP2A dephosphorylates BAK1, signaling from BAK1-FLS2 is decreased. Reverse genetics and biochemical experiments indicated that a PP2A complex containing C4, A1, and B’η or B’ζ dephosphorylates BAK1 [56]. The B subunits B’η and B’ζ had been selected for the experiments in the first place based on increased expression of these genes in response to biotic stress. The experiments confirmed that these subunits took part in dephosphorylation of BAK1; however, the involvement of other B’ subunits in this process is not ruled out. Another PRR receptor also uses BAK1 as a co-receptor, but this EFR (EF-TU RECEPTOR) has thus far not been investigated in connection with PP2A. 

B’γ, which is 80% identical with B’ζ on the protein level, was identified as necessary for preventing unnecessary defense responses [33]. The *b’γ* mutant showed yellowing spots even in the absence of pathogens, and the mutant also had higher levels of ROS (reactive oxygen species), a hallmark of PTI and ETI (pattern-triggered immunity and effector-triggered immunity) [57]. Both Arabidopsis *b’γ* and *b’θ* mutants were shown to have increased resistance to *Pseudomonas syringae* pv. tomato DC3000 proliferation [58]. Evidence was also found that B’γ interferes with activity and transcript abundance of a calcium-regulated protein kinase, i.e., CPK1, and thereby affected resistance to the fungus *Botrytis cinerea* [53]. In wheat, Kang and co-workers identified a *B’* gene as linked with resistance to *Blumeria graminis* (powdery mildew), by genome-wide association mapping [59]. Downstream of PRRs are NLRs (NUCLEOTIDE-BINDING LEUCINE-RICH REPEAT RECEPTORS), MAPKs (MITOGEN-ACTIVATED PROTEIN KINASES), CDPKs (CALCIUM-DEPENDENT PROTEIN KINASES), and calcium channels (including resistosomes) [57]. The NLRs, MAPKs, and CDPKs are all regulated by phosphorylation, and various protein phosphatases act as antagonists to the protein kinases to regulate signaling. It should be mentioned that many different protein phosphatases, not only PP2A enzymes, are involved, but substrates for different protein phosphatases are still scarcely identified. 

To fight back the plant’s defense system, bacteria produce so-called effectors. AvrE-family Type III effector proteins (T3Es) are present in *Pseudomonas*, *Pantoea*, *Ralstonia*, *Erwinia*, *Dickeya*, and *Pectobacterium*, and both dicots and monocots are attacked [60,61]. Yeast-two-hybrid assays, immunoprecipitation, and BiFC showed that T3Es (fragments) interacted with B’ in maize and Arabidopsis. Examination of Arabidopsis showed T3Es interactions with B’α and B’β, and only weak interactions with B’κ, but no interaction with B’η. Jin and collaborators [60] also tested Arabidopsis *pp2a* mutants for effects of the bacterial protein flg22 (flaggelin22) on the induction of ROS and confirmed previous results [56] that flg22 induced high ROS production in *b’η* and *b’ζ* mutants. Additionally, they found high ROS in *b’θ* (not in the six other *b’* mutants tested; *b’κ* and *b’ε* were not tested). Altogether, there is strong evidence that B’ subunits, specifically γ, η, ζ, and θ, are important in defense reactions. The B’ subunits were associated with different cellular localization. B’η, ζ, and possibly θ appeared to target the signaling pathways at the level of signal perception at the plasmalemma, while B’α and β would interact with T3Es effectors, which are likely to be found in association with the nucleus [61]. 

In recent years, the awareness of health-bringing microbes in humans as well as plants has become highly relevant [62,63]. When Arabidopsis is grown in Petri dishes with plant growth promoting bacteria (PGPB), for instance *Pseudomonas simiae* or *Azospirillum brasiliense*, roots become shorter, thicker, more branched, and shoot weight increases [64]. WT and *pp2a* mutants grown in the presence of PGPB were studied and revealed that the catalytic subunits double mutant *pp2a*-*c2* × *pp2a*-*c5* did not show all the positive effects on growth, indicating that the catalytic subunits belonging to subgroup 1 were involved in the response to plant growth promoting bacteria. Additionally, the experiments by Averkina and collaborators [64] indicated that the *b’θ* mutant responded different from WT. Altogether, the B’ subunits are clearly involved in interactions between microbes and plants, and interact with BAK1, CPK1, and type III effectors. Further investigations should reveal more targets and specifications on how the regulatory B’ subunits are involved.

## 6. Transition to Reproductive Phase—Five *b’* Single Mutants (*γ, ζ, η, θ, κ*) Are Late Flowering

The transition from a vegetative phase to a flowering phase is crucial for completion of the life cycle in plants. This is a process influenced by both endogenous components such as aging and environmental components such as nutrition, light, temperature, and various stresses. Historically, four basic flowering regulatory pathways were suggested and referred to as the autonomous, vernalization, photoperiod, and gibberellin pathway. Specific groups of genes were ascribed to the different pathways. The autonomous pathway and vernalization pathway were directed towards an inhibitor of flowering, *FLC* (*FLOWERING LOCUS C*). Activation of the autonomous or vernalization pathways led to inhibition of the inhibitor, and then release of the activators *FT* (*FLOWERING LOCUS T*) and *SOC1* (*SUPPRESSOR OF OVEREXPRESSION OF CONSTANS 1*) (Figure 1). In recent years, about 300 genes that influence flowering transitions have been identified, including microRNA genes and genes leading to changes in chromatin structure. For Arabidopsis, seven different signaling pathways promoting flowering transition are now defined (reviewed in [65,66]). Depending on the PP2A subunit in question, PP2A is an important component in the flowering process both as a positive and negative regulator. Some regulatory subunits appear also to be indifferent to flowering because *b’α* and *b’β* single mutants behaved similar to WT. All other *b’* mutants tested, i.e., *b’γ* [67], *b’ζ* [68], *b’η* [58], *b’θ*, [24], and *b’κ* (B. Heidari, unpublished), were late flowering. Interestingly, the different non-related group of PP2A regulators, i.e., B55 subunits, had the opposite effects, and *b55α* and *b55β* were early flowering. Hence, *B’* are positive regulators of flowering, whereas *B55* are negative regulators. Experiments testing gene expression and effects of long days and short days indicated that B’ were positive regulators in the autonomous pathway and B55 were negative regulators, possibly further downstream [67] (Figure 1). The functions and targets of B’ in different flowering pathways (autonomous pathway, photoperiodic pathway, vernalization pathway, age pathway, gibberellin pathway, and ambient temperature pathway) are, nevertheless, still poorly understood. While B’ single mutants only show altered expression levels of *FLC*, several of the other proteins that play key roles in other pathways to flowering integrator genes also carry potential motifs for B’ interactions, indicating a likely function of B’ in other flowering pathways (Table 2). Therefore, the possibility that B’ subunits have redundant functions in flowering pathways other than the autonomous pathway cannot be formally ruled out. The study of flowering time in B’ multiplex mutants will be of high interest in future research to overcome the hurdle of functional redundancy among B’ subunits.

## 7. Cell Division

B’ subunits also play an important and inimitable role during different phases of cell division, both in mitosis and meiosis. In *S. cerevisiae*, the B’ subunit (RTS1) plays a role in chromosome bi-orientation, preventing premature loss of centromeric cohesion in meiosis I and reorganization of septins during cytokinesis. Furthermore, in *S. pombe*, B’ subunits (Par1 and Par2) were shown to control mitotic exit via a relay mechanism consisting of PP1 and PP2A. The principle of such a phosphatase relay, in which recruitment of PP1 causes activation of PP2A to control mitosis, is also conserved in mammals [28]. In mammals, B’ is also known to interact with SHUGOSHIN at the centromeres and with BUBR1 (a pseudokinase) at the outer layer of kinetochores, where it counterbalances premature chromatid segregation and entry into anaphase. In Arabidopsis, the B’δ is likely involved in cell division, because B’δ relocates to the phragmoplast during mitosis [34]. It should also be mentioned that TON2/FASS is a much-studied B’’ subunit in Arabidopsis involved in the organization of microtubules in dividing as well as non-dividing cells [69]. In this section, we summarize the current state of knowledge and possible future research directions regarding the role B’ in plant cell division.

### 7.1. B’α and B’β Bind Shugoshin and Are Needed for Proper Meiosis in Plants

Before meiosis, in which four haploid cells arise from the original diploid cell, the DNA in each chromosome has been duplicated during S phase so that each chromosome consists of two identical sister chromatids. After exchange of DNA between pairs of homologous chromosomes, the chromosomes are lined up along the metaphase plate. Spindle fibers attached to each chromosome then separate the homologous chromosomes apart towards different poles in the cell, unlike mitosis where the sister chromatids separate. After this follows the second part of meiosis, where the sister chromatids are pulled apart much like in mitosis. Cell division then results in four haploid cells. Special proteins form a cohesin complex, which is conserved in eukaryotes and keeps sister chromatids together during the first part of meiosis. The REC8/SYN1 (RECOMBINATION DEFECTIVE 8/SYNAPTIC1) is part of this complex, and phosphorylation of REC8 initiates reactions leading to loss of chromatid cohesion [70]. The stepwise release of sister chromosomes during meiosis is essential for the equal distribution of genes to the daughter cells. Recently, it was shown that in an Arabidopsis double mutant *b’αβ* separation of chromosomes in anaphase I of meiosis was premature [32,50]. B’α or B’β localize to the centromeric region where the PP2A complex keeps REC8/SYN1 dephosphorylated for stability of the cohesin complex that assures cohesion of chromatids. Using yeast-two-hybrid and BiFC assays, Zhang and co-workers found interaction of B’α and B’β also with other cohesin-associated proteins, SGO1 and SGO2 (SHUGOSHIN 1 and 2), whereas other B’ subunits (ζ, ε, θ, γ) did not interact with SGO1 and 2 [50]. Aberrant separation of chromatids in the *b’αβ* double mutant led to unbalanced chromosome delivery to the four haploid daughter cells during meiosis. In contrast to 97% normal haploids in the wild type, only about 31% of the poles in *b’αβ* obtained five chromosomes and could develop into normal gametes. The remaining 69% were aneuploid which usually led to premature abortion of the gametes. Both male and female sterility was observed in the descendants of self-fertilized *b’αβ* double mutants [50]. Arabidopsis B’α and B’β have repeatedly been shown to be localized to the nucleus and to some degree to the cytosol [26,32,50] Apparently, B’α and B’β are both involved in functions in the nucleus by assuring correct meiosis and by being involved in dephosphorylation of transcription factors.

### 7.2. A Dephosphorylation Relay with PP1, PP2A-B55, and PP2A-B56 Controlling Mitosis Exit May Be Conserved in Plants

In all eukaryotes, the cell cycle consists of four stages (G1, S, G2, M) with two major checkpoints, at the transition from G1 to S stage and from G2 to M stage (Figure 2). Protein kinases and phosphatases are essential for control of the cell cycle, and the basic principle is that protein kinases push the cell cycle forward while protein phosphatases slow the cell cycle, although this is simplified and exception are easily pointed out. As in other eukaryotes, plants rely on CDKs (CYCLIN DEPENDENT KINASES) in complex with activator proteins, cyclins, to bring the cell cycle forward. Plants have an especially large number of cyclins, more than 50 homologs in Arabidopsis. CDKAs and CDKBs are crucial in promoting the cell cycle. CDKAs are active throughout the cell cycle and are orthologs of Cdc2 in *S. pombe* and CDK1 in human. CDKBs are plant specific and active in G2 and M phase [71].

At the beginning of G1, after exiting mitosis, CDKs and a master regulator RB (RETINOBLASTSOMA) in animals and RBR (RB RELATED) in plants are in a hypophosphorylated state [4,72,73]. In mammals, PP1 and PP2A have long been considered the most important protein phosphatase for dephosphorylating RB, as reviewed by Wlodarchak and Xing [4]. Both PP2A-B55 and PP2A-B’’ can dephosphorylate RB in mammals. Dephosphorylation of RBR in synchronized rice cell suspension cultures showed that B’’ interacted with RBR, and inhibition studies with variable okadaic acid concentrations pointed to PP1 also being important for dephosphorylation of RBR [73]. Thus far, there have been no reports of B’ being involved in dephosphorylation of RBR. In its hypophosphorylated state, RB(R) is a transcription repressor, but after phosphorylation on multiple sites by CDKs repression is released, necessary transcription becomes active, and the cell cycle can proceed into S phase. In S-phase, DNA is synthesized, which is regulated by a protein complex CDC6 (CELL DIVISION CONTROL 6) that is known to be controlled by PP2A-B’’ in mammals [4]. In all eukaryotes, high activity of CDKs (CDKB and A in plants) is crucial for cells to enter mitosis, and complex feedback loops control the activity of CDKs. Phosphorylation of a large group of CDK substrates takes place, which activates proteins for participation in mitosis. In mammals, CDK2 is important for entry, and regulation of the mammalian CDK2 relies on dephosphorylation by CDC25 (activating CDK2) and phosphorylation by WEE1 (inactivating CDK2). In plants, orthologs of CDC25 are absent and orthologs of WEE1 function differently [71,74]. Other mitosis regulators, such as ENSA/ARPP-19, are conserved in plants, [75] and are apparently involved in keeping PP2A-B55 inactive at the start of mitosis [71]. In mammals, B’ is also involved in regulation of mitotic entrance [76], but this has, thus far, not been studied in plants. PP2A-B’ performs crucial tasks during mitosis by interactions with shugoshin to preserve binding between sister chromatids until the cell is ready for separation of chromatids. Which B’ would be involved in regulating chromatid binding during mitosis has not been resolved in plants, but in contrast to chromatid binding during meiosis, B’α and B’β do not appear to be involved [32]. To exit mitosis, PP2A-B55 needs to be activated for dephosphorylation of substrates that had been phosphorylated by CDKs [71]. As mitosis proceeds, ENSA/ARPP-19 becomes non-phosphorylated and no longer inhibits B55, which can then engage to antagonize CDKs [4,76]. PP1 initiates the activation of PP2A-B55 by promoting dephosphorylation, and thereby, inactivation of the inhibitor ENSA/ARPP-19. PP1 may also dephosphorylate and activate B55 directly [28,76]. B55 has a docking site (K/RxVxF motif) for PP1 that is conserved in *S. pombe*, *S. cerevisiae*, and animals [28], and also conserved in plants, as presented here (Table 3). Grallert and co-workers have pointed out a phosphatase relay, where PP1, PP2A-B55, and PP2A-B’ are linked and are all repressed when the cell enters mitosis, whereafter PP1 is later activated and induces dephosphorylation and activation of PP2A-B55 and PP2A-B’. In Arabidopsis (Table 3), both B55 subunits and seven of the nine B’ subunits have a K/RxVxF docking motif for PP1. In addition, all seven B’ subunits bear a secondary PP1 docking motif, G/SILK/R as found also in yeast and animals. Five Arabidopsis B’ subunits also have a putative POLO kinase phosphorylation site between the G/SILK/R and K/RxVxF motifs, as in yeast. Animal B’ has a conserved Aurora consensus at this site. In *S. pombe*, B’-S378 was confirmed to be phosphorylated by Polo kinase, and dephosphorylated by the help of PP1 and PP2A-B55 (Grallert et al. 2015). The conservation of the docking motifs for PP1 in both B55 and B’, and consensus of POLO kinase motif in B’ suggests conservation of a PP1-B55-B’ dephosphorelay in plants (Table 3). 

## 8. Conclusions

The B’α and B’β subunits have overlapping functions in regulating chromatid cohesion during meiosis. The third member of this clade, i.e., B’ε, has not been much studied, but there were no indications that B’ε could replace B’α and B’β in meiosis [32]. B’ε could possibly have evolved to engage in chromatid cohesion during mitosis, but this has, thus far, not been investigated and is an open question in further investigation regarding B’ε. Reverse genetics showed that in transition to flowering, B’α and B’β were neutral, whereas all other B’ tested (γ, ζ, η, θ, and κ) acted positively on transition to flowering. Interestingly, the five B’ subunits important for promoting flowering in Arabidopsis coincide with the B’ subunits containing three conserved motifs putatively important for a phosphatase relay in mitosis. This supports the view that a PP1-PP2A dephosphorelay can be important in a variety of signaling contexts throughout eukaryotes [28]. The transient interaction between PP2A and its substrates has been a major obstacle to progress in protein phosphatase research, but new methods for labeling interacting proteins by help of an efficient biotin ligase linked with PP2A is promising for further progress [77]. Such methods can be important for clarifying the roles of different PP2A subunits, including B’ subunits, in the cell cycle.

## Figures and Tables

**Figure 1 ijms-24-12255-f001:**
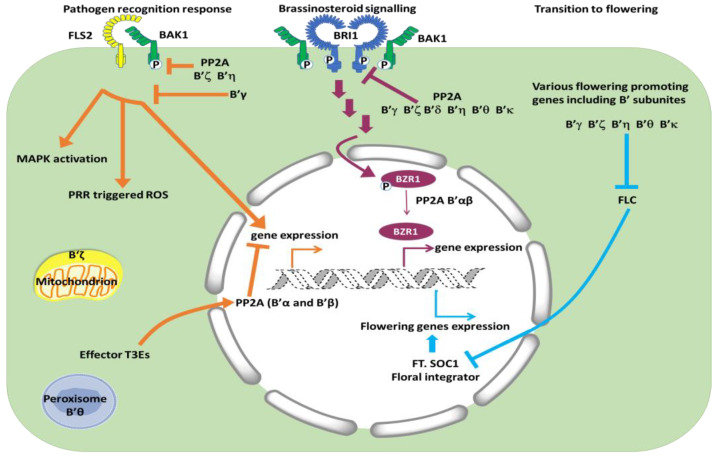
Simplified scheme of the best-established functions and subcellular localization of B’ subunits. Pathways to the left: PP2A complexes with B’ζ and η dephosphorylate the co-receptor BAK1 and act as negative regulators of signaling from flagellin recognized by FLS2. B’θ and B’ζ have been found associated with peroxisomes and mitochondria, respectively. Pathways in the middle: PP2A complexes with B’γ, ζ, δ, η, θ, κ dephosphorylate and inactivate the BR receptor BRI1 at the plasma lemma, whereas PP2A complexes with B’α, β dephosphorylate transcription factor BZR1 and keeps active BZR1 in the nucleus, thereby promoting BR signaling. Pathways to the right: Same B’ subunits as act negatively in BR signaling promote flowering (B’δ not tested), probably through inhibition of the FLC inhibitor, though interacting proteins have not been identified for this process.

**Figure 2 ijms-24-12255-f002:**
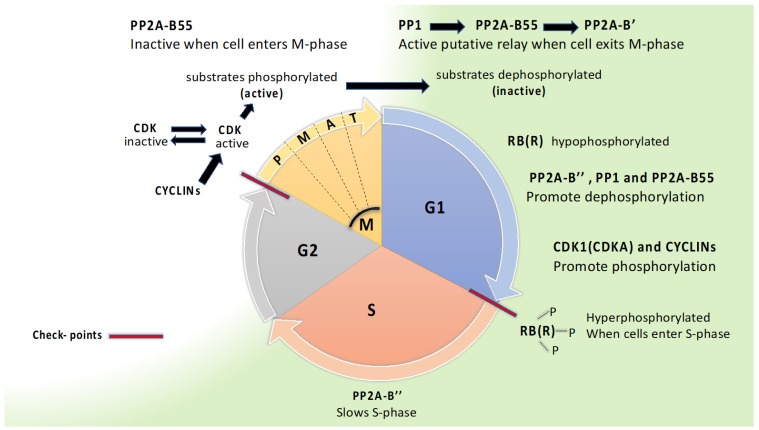
PP2A engagement in the cell cycle. The cell cycle is divided into G1 (GAP1 phase), S (synthesis of DNA phase), G2 (GAP2 phase), and M (mitosis). Mitosis is divided into P (prophase), M (metaphase), A (anaphase), and T (telophase). The two major checkpoints are indicated with red lines. The RB(R) (RETINABLASTOMA (RELATED)) protein is conserved in eukaryotes, and phosphorylation of RB(R) by CDKs is crucial for G1 to S transition, whereas dephosphorylation keeps cells in G1. Activation of CDK2 in mammals (CDKB, A in Arabidopsis) and inactivation of PP2A-55 is crucial for bringing cells from G2 to M. To exit M, a dephosphorelay where PP1 activates PP2A-B55 and PP2A-B’ is found in fission yeast, and possibly conserved across eukaryotes.

**Table 2 ijms-24-12255-t002:** Potential docking motif for B’ on Arabidopsis marker proteins of various flowering pathways.

Flowering Time Marker Genes	AGI Number	Flowering Pathway	B’56 Docking Motif
FLK (FLOWERING LOCUS K)	At3g04610	Autonomous	IKKIVEETR
FLD (FLOWERING LOCUS D)	At3g10390	Autonomous	LRGIYEPQG LYYLCETLG
FLC (FLOWERING LOCUS C)	At5g10140	Autonomous	LVQLEEHLE
SVP (SHORT VEGETATIVE PHASE)	At2g22540	Ambient temperature	MSEISELQK LTRVIETKS
SPL3 (SQUAMOSA PROMOTER BINDING PROTEIN-LIKE3)	At2g33810	Age	LRELSEEEE
GI (GIGANTEA)	At1g22770	Photoperiodic	LLGLLEAPP ICTIWEAAY LLKVLEYLP ILAILEALF
FKF1 (FLAVIN-BINDING KELCH REPEAT F-BOX 1)	At1g68050	Photoperiodic	LNELHELCL
VRN2 (VERNALIZATION 2)	At4g16845	Vernalization	CNTILENCR
MYB33 (MYB DOMAIN PROTEIN 33)	At5g06100	Gibberellin	LGIVKETGS

**Table 3 ijms-24-12255-t003:** Conserved PP1 and POLO kinase docking motifs on B55 and B’ subunits. The K/RxVxF (marked yellow) docking motif for PP1 is present in both B’ and B55 subunits. The G/SILK/R (marked green) PP1 docking motif is present in B’ subunits. A POLO kinase motif is highlighted in red.

Subunit	AGI Number	PP1 Docking Motifs and POLO Kinase Consensus
B’α	At5g03470	TVIRGLLKFWPVTNCTKEVLFLGELEEVLE
B’ε	At3g54930	TVIRGLLKFWPLTNCQKEVLFLGELEEVLD
B’γ	At4g15415	TVIRGLLKYWPVTNSSKEVMFLGELEEVLE
B’ζ	At3g21650	TVIRGLLKYWPVTNSSKEVMFLGELEEVLE
B’η	At3g26020	TVIRGLLKYWPVTNSSKEVMFLNELEEVVI
B’θ	At1g13460	TVIRGLLKSWPVTNSSKEVMFLNELEEVLE
B’κ	At5g25510	VVIKGLLKFWPITNSQKEVMFLGEVEEIVE
B55α	At1g51690	RERERGNHLATGDRGGRVVLFERTDTNNSS
B55β	At1g17720	EFDKSGDHLATGDRGGRVVLFERTDTKDHG

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
