# Peer review of "Distinct Clades of Protein Phosphatase 2A Regulatory B’/B56 Subunits Engage in Different Physiological Processes"

_ijms, 2023, doi:10.3390/ijms241512255_

Round 1

Reviewer 1 Report

I congratulate the authors for their review work and manuscript. Manuscript ID: ijms-2507578 "Distinct Clades of Protein Phosphatase 2A Regulatory B’/B56 Subunits Engage in Different Physiological Processes" by Heidari and collaborators covers the vast literature on protein dephosphorylation over the last couple of decades. A lot of progress has been made in understanding the complexity of phosphatase functions in plants, and this review article provides a good depth of detail and the different areas that have sought the most progress. The text is well written and figures and tables included are helpful. I believe the detailed descriptions of functions and cellular localizations of the different groups will be interesting to the readers of IJMS, however, more illustrations of important concepts would aid the reader to grasp all this complexity. For example additional figures exemplifying the general structural organization of the phosphatase holoenzyme with its different subunits, the interaction and mechanism of action of specific subtypes and their metabolic effects explored in each text section, sequence alignments highlighting important protein motifs, and general schemes summarizing the metabolic consequences of their actions would enrich the manuscript even further. This would also help the manuscript be more accessed and cited.
I have no issues with the current version and I will still recommend the manuscript be accepted for publication, however I'm giving the authors an opportunity to add more figures as a strategy to engage more readers and consequently increase its citations.

Reviewer 2 Report

   The manuscript by Heidari et al. is devoted to reviewing role of distinct clades of protein phosphatase 2A regulatory B’/B56 subunits engage in different physiological processes. This manuscript seems to be interesting and useful. I have only minor remarks.

   1. Additional figure summarizing physiological roles of PP2A can make manuscript more suitable for potential readers.

   2. P. 4, lines 167-168: “Reversible protein phosphorylation is essential for regulation of various processes also in chloroplasts and mitochondria”. Please clarify these processes. Does PP2A participate in photosynthetic regulation? In state transition of LHCII? I suppose that description of these points can additionally improve manuscript.

Reviewer 3 Report

Please find the attachment with the reviewed version of the manuscript including minor corrections.

The manuscript "Distinct Clades of Protein Phosphatase 2A Regulatory B'/B56 Subunits Engage in Different Physiological Processes" is well-written, clear, and compelling. However, the only chapter that requires corrections is the conclusions. In the present form, the conclusions are more like a continuation of the informative chapter than an objective summary with a few perspectives. 

The manuscript will be interesting for IJMS readers.

Minor editing of English language is required.
